# LEVERAGING HUMAN COLLABORATION: LEARNING FROM HUMAN-VERIFIED LABELS

## ABSTRACT

Pre-trained Vision-Language Models (VLMs) exhibit strong zero-shot recognition accuracy, making them widely used for generating labels. However, the labels generated by VLMs are often noisy and lacking human collaboration, leading to significant performance degradation for learning classifiers. To address this challenge, as shown in Figure 1, we propose a novel setting, called Human-Verified Labels (HVLs), to verify whether the labels generated by VLMs are correct with human collaboration. Specifically, HVLs enhance the quality of the labels by incorporating human verification for each label, which only needs limited labor costs. Besides, we propose a risk-consistent estimator to explore and leverage the underlying correlations between VLM-generated and human verification labels. Experimental results demonstrate the effectiveness of the proposed HVL setting.

## 1 INTRODUCTION

Recently, pre-trained Vision-Language Models (VLMs) (Radford et al., 2021; Liu et al., 2023) trained on massive amounts of image-text data have exhibited revolutionary cross-modal understanding capabilities. In particular, they can serve as efficient label generators, significantly reducing annotation costs and providing a valuable source of weak supervision, reducing the reliance on manually annotated data (Mirza et al., 2023). To better leverage these VLM-generated labels as weak supervision, several recent studies have proposed improved strategies (Zhang et al., 2024; Li et al., 2024). For instance, Zhang et al. (2024) introduce a method that refines candidate pseudolabels through confidence-guided intra- and inter-instance selection, enabling more effective fine-tuning of VLMs with unlabeled data. Similarly, Li et al. (2024) propose a weakly supervised labeling approach based on true-false labels, allowing VLMs to verify whether an instance aligns with a sampled label and to learn through a risk-consistent framework.

Although VLMs can significantly reduce annotation costs for massive unlabeled datasets (Menghini et al., 2023), this automated labeling method often generates noisy labels (Wang et al., 2022; Menghini et al., 2023). As illustrated in Figure 1, an image with the ground-truth label "Trout" may be incorrectly labeled as "Flatfish". Training directly on such noisy labels can severely degrade classifier performance, thereby limiting the applicability of VLMs. Unfortunately, most existing methods rely solely on labels generated by VLMs throughout both the annotation and training stages, with no human verification to mitigate label noise.

In this paper, we introduce Human-Verified Labels (HVLs), a novel annotation setting that effectively distinguishes noisy labels by incorporating limited human collaboration. The HVLs indicate whether a label generated by VLMs aligns with the ground-truth label based on human verification. Specifically, each instance is assigned a "True" or "False" label based on this human verification process. As illustrated in Figure 1, for an image with the ground-truth label "Trout", human annotators can readily identify the erroneous label "Flatfish" and mark it as "False". Conversely, they assign a "True" label when the label generated by VLMs aligns with the ground-truth. HVLs require only limited human effort to verify labels generated by VLMs, yet consistently achieve high-quality annotation information, representing a preliminary exploration of a new human-AI collaborative labeling mechanism.

To effectively learn from HVLs data, we first theoretically derive a risk-consistent estimator to leverage the underlying correlations between the probability distribution of labels generated by VLMs and human verification results. Building on this, to further improve the accuracy of conditional

Figure 1: Generation process of the Human-Verified Labels (HVLs), combining labels generated by VLM with human verification. The results shown are based on VLMs such as CLIP. The example images are from the CIFAR-100. HLVs significantly improve the quality of annotations.

probability in the proposed estimator, we propose a hybrid probability estimation method that jointly leverages per-sample knowledge from both VLMs and the learning model to enhance conditional probability estimation accuracy. Our main contributions are summarized as follows:

- We propose a novel data annotation setting that effectively integrates limited human verification with labels generated by VLMs, achieving both high-quality annotation information and improved labeling efficiency.

- We propose a risk-consistent estimator to explore and leverage human verification information in HVLs. In addition, we introduce a hybrid probability estimation method that combines knowledge from both VLMs and the learning model to achieve more accurate conditional probability estimation.

- We demonstrate through extensive experiments that our method outperforms the state-of-the-art methods.

## 2 METHOD

In this section, we present a risk-consistent method for learning from Human-Verified Labels (HVLs). We begin with the problem formalization and the labeling process of HVLs, followed by a theoretical derivation of a risk-consistent estimator.

### 2.1 PROBLEM FORMALIZATION

**Multi-Class Classification.** Let $\mathcal{X} \in \mathbb{R}^d$ denote the feature space and $\mathcal{Y} = \{1, \ldots, K\}$ denote the label space, where $d$ is the feature space dimension and $K$ is the number of classes. Consider a labeled dataset $\mathcal{D} = \{(x_l, y_l)\}_{l=1}^N$, where each instance $x_l \in \mathcal{X}$ is associated with a ground-truth label $y_l \in \mathcal{Y}$, and $N$ is the total number of training samples. We assume that the samples are independently drawn from an unknown joint distribution $p(x, y)$. Ordinary multi-class classification aims to learn a classifier $f(x) : \mathcal{X} \to \mathcal{Y}$ by minimizing the expected classification risk as follows:

$$R_M(f) = \mathbb{E}_{x \sim p(x)} \sum_{i=1}^K p(y = i \mid x) \, \mathcal{L}(f(x), i), \tag{1}$$

where $\mathcal{L}(f(x), y)$ denotes the multi-class loss function, $p(x)$ is the probability distribution of the input $x$, and $\mathbb{E}_{x \sim p(x)}$ is the expectation over the density $p(x)$.

**Human-Verified Labels.** In this paper, the human-verified labeled training dataset is denoted as $\mathcal{D}_{\mathrm{HVL}} = \{(x_l, Y_l)\}_{l=1}^N$, where each sample is drawn randomly and uniformly from an unknown distribution with density $p(x, Y)$. Each HVL is represented as $Y_l = [\hat{y}_l, h_l]$, where $\hat{y}_l \in \mathcal{Y}$ is the label generated by the VLM, and the human annotator verifies whether the label $\hat{y}_l$ is correct, indicated by $h_l \in \{0, 1\}$, a binary label reflecting the verification result. Specifically, $h_l = 1$ means that the instance $x_l$ belongs to the category $\hat{y}_l$, while $h_l = 0$ indicates that it does not. The goal is to learn a classifier $f(x)$ from this human-verified labeled dataset, such that it can accurately classify previously unseen images.

Compared to ordinary multi-class labels, the Human-Verified Labels (HVLs) consist of two components: ① VLM annotation and ② human verification. As illustrated in Figure 1, consider an image with ground truth "Trout" and candidate label set {"Apple", "Aquarium Fish", "Baby", "Bear", "Beaver", ..., "Trout", ... }. We first utilize VLM, such as CLIP or Qwen-VL, to generate an initial label for the image. Denote this predicted label as $\hat{y}$, which corresponds to a category index $i$ (e.g., the class "Flatfish"). We then introduce a novel human collaboration mechanism, in which human annotator verifies the correctness of $\hat{y}$. Unlike traditional annotation that requires labeling from scratch, annotators only need to judge whether $\hat{y}$ is correct for the given image. If deemed incorrect, a binary label $h = 0$ is assigned, meaning the label generated by VLM is wrong. As a result, the sample is finally annotated with a HVL represented as $Y = [i, 0]$.

**Superiority of HVLs.** The proposed approach enhances annotation quality by verifying labels generated by VLMs, which significantly reduces annotation costs for large-scale annotation tasks. As shown in Figure 1, directly using VLM-generated labels for training results in limited performance. In contrast, HVLs greatly improve performance, increasing accuracy by 41.00%, 12.55%, and 11.41% on the EuroSAT, Stanford Cars, and DTD datasets, respectively. Furthermore, HVLs prevent models from learning erroneous patterns by establishing a formal framework to leverage the interplay between VLM-generated labels and human verification.

Although HVLs do not directly correct the labels, they provide the classifier with noise-resistant annotation signals through human verification by distinguishing between correct and incorrect data. This annotation strategy effectively delineates a human-verified "error zone" within the training data, and compared to traditional VLM annotation, it delivers higher-quality supervision signals with only limited labor cost.

## 2.2 Risk-Consistent Estimator

In this section, based on the previously defined problem, we propose a risk-consistent method (Feng et al., 2020; Xu et al., 2022). To facilitate theoretical analysis, we assume that the ground-truth labels and the HVLs satisfy the following condition.

**Definition 1 (HVLs Condition).** Given a labeled sample $(x, Y) \in \mathcal{D}_{\mathrm{HVL}}$, $\forall i, j \in \mathcal{Y}$, we say $Y = [i, h]$ is a HVL if it satisfies the following condition:

$$p(y = i \mid Y = [i, 1], x) = 1 \tag{2}$$

$$\sum_{i \neq j} p(y = i \mid Y = [j, 1], x) = 0. \tag{3}$$

Intuitively, this condition ensures that human verification can perfectly identify the labels generated by VLMs. Once a label is verified as true (i.e., $Y = [i, 1]$), the probability that it aligns with the ground-truth label is 1 (Eq. (2)). Eq. (3) further rules out all other possibilities: if the label $j$ in $Y = [j, 1]$ is verified as correct, then the true label must be $j$ and cannot be any other class.

Since the conditional probability $p(y = i \mid x)$ in Eq. (1) cannot be directly obtained, we introduce the following Lemma.

**Lemma 2.** Under the HVLs Condition, the conditional probabilities $p(y = i \mid x)$ can be expressed as a combination of the probability of human-verified true labels and the corrected probabilities of human-verified false labels as follows:

$$p(y{=}i \mid x) = \underbrace{p(Y{=}[i, 1] \mid x)}_{\text{Human-Verified True Labels}} + \underbrace{\sum_{\substack{j=1 \\ j \neq i}}^{K} p(y{=}i \mid Y{=}[j, 0], x)\, p(Y{=}[j, 0] \mid x)}_{\text{Human-Verified False Labels}} \tag{4}$$

Essentially, this formulation decomposes the conditional probability of the ground truth label into two parts: the probability of human-verified true labels, and a weighted probability derived from the human-verified false labels. This decomposition enables the model to rely solely on observable HVLs information, without requiring access to ground-truth labels, while still achieving a reasonable estimation of the true distribution. The proof is provided in the Appendix B.1.

**Theorem 3.** To deal with the learning problem of HVLs, according to Condition 1 and Lemma 2, the classification risk $R_M(f)$ in Eq. (1) could be rewritten as:

$$R_{\text{HVL}}(f) = \mathbb{E}_{p(x, Y=[\hat{y}, 0])} \, \bar{\mathcal{L}}(f(x), Y) + \mathbb{E}_{p(x, Y=[\hat{y}, 1])} \, \mathcal{L}(f(x), Y) \tag{5}$$

This classification risk is divided into two parts: when the label is verified as true ($Y = [\hat{y}, 1]$) by human, the standard loss $\mathcal{L}(f(x), Y)$ is directly applied to the verified class; when the label is verified as false ($Y = [\hat{y}, 0]$), the loss is redefined as a weighted sum over other classes, where the weights are given by the predicted probabilities of the sample belonging to those classes, defined as

$$\bar{\mathcal{L}}(f(x), Y) = \sum_{\substack{i=1 \\ i \neq j}}^{K} p(y{=}i \mid Y{=}[j, 0], x) \, \mathcal{L}(f(x), Y) \tag{6}$$

Together, they enable learning directly from HVLs without relying on ground-truth labels. The proof is provided in the Appendix B.2.

To facilitate our subsequent formulation, we explicitly reorganize the HVL dataset into two disjoint subsets: $\mathcal{D}_{\text{HVL}_T}{=}\{(x_i, Y_i{=}[\hat{y}, 1])\}_{i=1}^{N_T}$ and $\mathcal{D}_{\text{HVL}_F}{=}\{(x_j, Y_j{=}[\hat{y}, 0])\}_{j=1}^{N_F}$, where $N_F$ and $N_T$ denote the number of instances in $\mathcal{D}_{\text{HVL}_T}$ and $\mathcal{D}_{\text{HVL}_F}$. Then the empirical risk estimator can be expressed as:

$$\hat{R}_{\text{HVL}}(f) = \frac{1}{N_F} \sum_{i=1}^{N_F} \bar{\mathcal{L}}(f(x_i), Y_i) + \frac{1}{N_T} \sum_{j=1}^{N_T} \mathcal{L}(f(x_j), Y_j) \tag{7}$$

Then, we can learn a multi-class classifier $f(x) : \mathcal{X} \to \mathcal{Y}$ by minimizing the proposed empirical approximation of the risk-consistent estimator in Eq. (7).

## 2.3 PRACTICAL IMPLEMENTATION

**Conditional Probability Estimation.** To compute and minimize the empirical risk estimator $\hat{R}_{\text{HVL}}$, we rely on accurate estimation of the conditional probability $p(y = i \mid Y{=}[j, 0], x)$. To this end, we propose a hybrid probability estimation method, which leverages the label distributions predicted by both the VLM and the current learning model to improve estimation accuracy.

Specifically, we first obtain the conditional probability distribution $\hat{P}$ using CLIP (Radford et al., 2021), which can be formulated as:

$$\hat{P} = Softmax\big(cos(x, \mathbb{Q}_T)\big) \tag{8}$$

where $\mathbb{Q}_T$ denotes the set of textual embeddings for all categories, and $x$ is the image embedding of the instance. Here, $cos(x, \mathbb{Q}_T)$ represents the cosine similarity between the image embedding and category embedding, which measures their directional alignment in the feature space. In addition to this, we also utilize the output of the learning model itself as another source of probability estimation:

$$\bar{P} = Softmax(f(x)) \tag{9}$$

To balance these two sources, we fuse them using a weighted sum to obtain the final estimated conditional probability distribution:

$$P = \lambda \cdot \hat{P} + (1 - \lambda) \cdot \bar{P} \tag{10}$$

where $\lambda \in (0, 1)$ is a hyperparameter that balances the static prior from CLIP ($P_{\text{CLIP}}$) and the dynamically updated model knowledge ($P_{\text{model}}$). While $P_{\text{CLIP}}$ provides stable global cues but may be biased, $P_{\text{model}}$ gradually aligns with the true distribution as training progresses. Thus, $\lambda$ helps combine their strengths for more stable and expressive probability estimation.

**Loss Functions.** In this paper, we adopt the widely used cross-entropy loss function for multi-class classification. For instances verified as correct by human annotators, we apply the standard

Table 1: Comparison results in terms of classification accuracy based on HVLs, which are constructed from labels generated by CLIP. The best accuracy (excluding fully supervised results) is highlighted in **bold**, and the second-best is underlined. HVL (CLIP) denotes the results using HVLs derived from CLIP-generated labels.

| | CIFAR-100 | Food-101 | Stanford Cars | Caltech-101 | DTD | EuroSAT | Average |
|---|---|---|---|---|---|---|---|
| **Accuracy of Labels Generated by CLIP** | | | | | | | |
| ZS CLIP (Radford et al., 2021) | 63.70 | 80.15 | 59.27 | 92.27 | 43.48 | 39.02 | 62.98 |
| **Supervised Labels Learning** | | | | | | | |
| FSFT | 84.41 | 84.99 | 81.43 | 96.71 | 71.45 | 96.83 | 85.97 |
| **Weakly Supervised Learning Methods** | | | | | | | |
| DIRK (Wu et al., 2024) | 77.21 | 77.54 | 66.78 | 93.52 | 57.16 | 79.44 | 75.28 |
| SPMI (Liu et al., 2024) | 70.91 | 72.95 | 59.13 | 94.85 | 61.45 | 80.99 | 73.38 |
| PaPi (Xia et al., 2023) | 79.14 | 79.84 | 64.07 | 95.27 | 57.13 | 80.61 | 76.01 |
| PLNL (Li et al., 2025) | 81.55 | 83.33 | 66.98 | 95.94 | 60.88 | 91.40 | 80.01 |
| CPL (Zhang et al., 2024) | 78.14 | 82.09 | 65.25 | 93.23 | 56.03 | 82.52 | 76.21 |
| TMP (Li et al., 2024) | 60.28 | 78.48 | 46.71 | 91.64 | 45.21 | 40.74 | 60.51 |
| **HVLs-based Methods** | | | | | | | |
| ZS LaFTer Mirza et al. (2023) | 73.08 | 80.25 | 56.55 | 93.06 | 50.59 | 71.04 | 70.76 |
| NLFT | 73.92 | 81.72 | 63.95 | 92.09 | 51.65 | 50.42 | 68.96 |
| TLFT | 81.46 | 83.80 | 73.03 | 95.38 | 60.82 | 82.16 | 79.44 |
| HVL (CLIP) | **82.67** | **84.23** | **76.50** | **96.39** | **63.06** | **91.42** | **82.38** |

cross-entropy loss directly. For instances marked as incorrect, we leverage the hybrid probability estimation proposed earlier to guide the model away from "error zone". This strategy effectively mitigates the negative impact of incorrect labels on training, thereby improving both performance and robustness.

**Model.** We adopt LaFTer (Mirza et al., 2023) as our base model and use the ViT-B/32-based CLIP (Radford et al., 2021) as the backbone network. Our method is applied by fine-tuning the LaFTer model that has been pre-trained on unlabeled data.

## 3 Experiments

### 3.1 Experimental Setup

**Dataset.** We evaluate our method on six multi-class image classification datasets, covering natural objects, fine-grained categories, and satellite imagery. Specifically, we use CIFAR-100 (Krizhevsky et al., 2009), Caltech-101 (Fei-Fei et al., 2004), and DTD (Cimpoi et al., 2014) for natural categories; Stanford Cars (Krause et al., 2013) and Food-101 (Bossard et al., 2014) for fine-grained classification; and EuroSAT (Helber et al., 2019) for satellite images. For all datasets, we replace the training labels with Human-Verified Labels (HVLs), while the test sets retain their ground-truth annotations. More information related to the datasets is shown in the Appendix B.3.

**Implementation Details.** In all our experiments, unless otherwise specified, we use the pretrained LaFTer model based on the CLIP model with a ViT-B/32 backbone, and fine-tune it on each dataset. The model is optimized using the AdamW optimizer (Loshchilov & Hutter, 2017) with an initial learning rate of 0.0005, and a StepLR scheduler. All models are trained for 50 epochs with a batch size of 50 on a single NVIDIA RTX 4090 GPU.

**Compared Methods.** To validate the effectiveness of our method, we compare it with a range of weakly supervised learning methods, including partial-label learning (Xia et al., 2023; Liu et al., 2024; Wu et al., 2024), complementary-label learning (Li et al., 2025), and automatic labeling learning method (Zhang et al., 2024; Li et al., 2024). Our method, Human-Verified Labels (HVLs), explicitly incorporates human verification to improve label quality.

We also explore three fine-tuning strategies based on HVLs to assess the influence of label reliability: (1) Noisy Label Fine-Tuning (NLFT), using all VLM-generated labels; (2)Trusted Label Fine-Tuning (TLFT), using only those verified as true by human; (3) Full Supervision Fine-Tuning (FSFT), using ground-truth labels. All methods are built upon the pretrained LaFTer model with the CLIP-ViT-B/32 backbone. Key statistics for the compared methods are summarized as follows:

Table 2: Comparison results in terms of classification accuracy based on HVLs, which are constructed from labels generated by Qwen. The best accuracy (excluding fully supervised results) is highlighted in **bold**, and the second-best is underlined. HVL (Qwen) denotes the results using HVLs derived from Qwen-generated labels.

| | CIFAR-100 | Food-101 | Stanford Cars | Caltech-101 | DTD | EuroSAT | Average |
|---|---|---|---|---|---|---|---|
| **Accuracy of Labels Generated by Qwen** | | | | | | | |
| ZS Qwen (Bai et al., 2025) | 46.57 | 53.21 | 44.30 | 84.16 | 57.62 | 32.19 | 53.01 |
| **Supervised Labels Learning** | | | | | | | |
| FSFT | 84.41 | 84.99 | 81.43 | 96.71 | 71.45 | 96.83 | 85.97 |
| **Weakly Supervised Learning Methods** | | | | | | | |
| DIRK (Wu et al., 2024) | 75.78 | 77.03 | 63.83 | 92.72 | 60.08 | 91.09 | 76.76 |
| SPMI (Liu et al., 2024) | 58.92 | 53.61 | 49.58 | 92.83 | 62.98 | 86.61 | 67.42 |
| PaPi (Xia et al., 2023) | 67.91 | 63.50 | 50.36 | 92.82 | 59.06 | 87.38 | 70.17 |
| PLNL (Li et al., 2025) | 77.26 | 76.59 | 54.71 | 93.96 | 65.25 | **94.12** | 76.98 |
| CPL (Zhang et al., 2024) | 78.14 | **82.09** | 65.25 | 93.23 | 56.03 | 82.52 | 76.21 |
| TMP (Li et al., 2024) | 61.09 | 78.43 | 46.98 | 91.72 | 45.33 | 41.09 | 60.77 |
| **HVLs-based Methods** | | | | | | | |
| ZS LaFTer Mirza et al. (2023) | 73.08 | 80.25 | 56.55 | 93.06 | 50.59 | 71.04 | 70.76 |
| NLFT | 57.68 | 58.83 | 43.58 | 92.09 | 54.72 | 58.73 | 60.94 |
| TLFT | 74.10 | 72.64 | 58.85 | 94.08 | 64.48 | 88.53 | 75.45 |
| HVL (Qwen) | **78.69** | 79.70 | **68.62** | **94.77** | **67.25** | 92.80 | **80.31** |

- DIRK (Wu et al., 2024), SPMI (Liu et al., 2024), PaPi (Xia et al., 2023): Partial-label methods trained with candidate label sets. We treat $Y = [\hat{y}, 1]$ samples as fully labeled, and for $Y = [\hat{y}, 0]$, construct candidate sets excluding the VLM-predicted class.

- PLNL (Li et al., 2025): A complementary-label method, where for $Y = [\hat{y}, 0]$ instances, the VLM-predicted class is treated as a negative class.

- CPL (Zhang et al., 2024) and TMP (Li et al., 2024): The automatic labeling learning method utilizes a pre-trained model for automated labeling and learning. The CPL method selects candidate label sets based on model confidence, and is trained directly on the original dataset without human supervision. It is worth noting that TMP cannot be used with the pre-trained LaFTer model due to its multi-modal prompt retrieving module. Therefore, we replaced the feature extractor with ViT-B/32 and conducted experiments using the model described in the original paper.

## 3.2 RESULTS OF HVLs GENERATED BY CLIP

In this section, we evaluate our method using HVLs generated through collaboration between VLM and human annotators. As shown in Table 1, our method outperforms most weakly supervised baselines, with up to 29.79% improvement on fine-grained datasets like Stanford Cars. This highlights the limitations of traditional methods in utilizing human verification information embedded in HVLs, and the effectiveness of our method in leveraging such signals.

Furthermore, compared to other HVL-based methods, ours achieves consistent gains, outperforming Zero-shot LaFTer and TLFT by 11.62% and 2.94% on average. While NLFT suffers from noisy labels, our method benefits even from labels verified as false by human, without needing manual correction. These results underscore the value of collaboration between VLMs and human annotators.

## 3.3 RESULTS OF HVLs GENERATED BY QWEN

To assess the impact of label quality, we replace CLIP with the dialogue-based model Qwen-VL-7B as the label generator and repeat the experiments. As shown in Table 2, Qwen produces lower zero-shot annotation accuracy than CLIP, leading to a significant drop in NLFT performance and confirming the sensitivity of supervised learning to label quality.

In contrast, HVL remains robust across all datasets, significantly outperforming weakly supervised baselines even when using noisier labels generated by Qwen. This highlights the effectiveness of HVL in mitigating the impact of noisy supervision and emphasizes the importance of incorporating human verification.

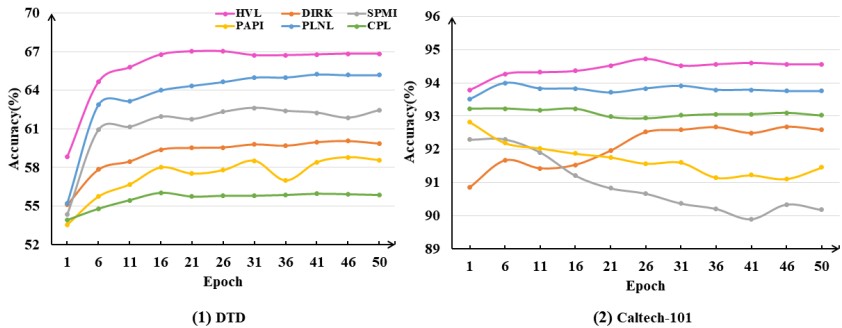

Figure 2: Convergence curves of different methods on the DTD and Caltech-101 datasets.

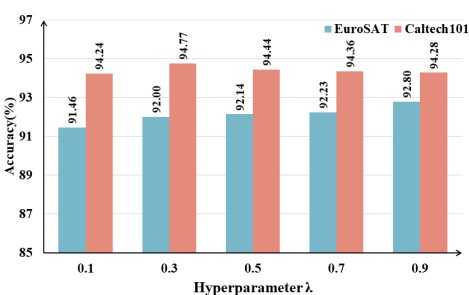

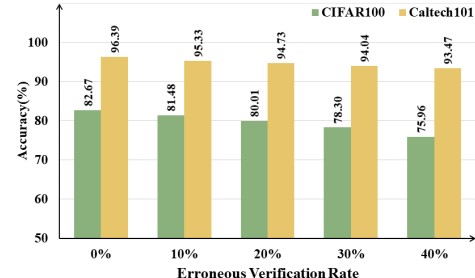

Figure 3: Experimental results on the influence of the hyperparameter $\lambda$. Experiments are performed on HVL data based labels generated by Qwen.

Figure 4: Experimental results on the influence of erroneous verification rate. Experiments are performed on HVL data based labels generated by CLIP.

Interestingly, on EuroSAT, HVL (Qwen) outperforms HVL (CLIP) by 1.38%, despite the overall lower annotation accuracy of Qwen. We attribute this to severe class imbalance in labels generated by CLIP on EuroSAT, which Qwen mitigates to some extent. Conversely, on Food-101, labels generated by Qwen are both inaccurate (53.21%) and biased, causing most methods to underperform, except CPL, which is trained on unlabeled data and avoids noisy labels entirely.

These findings suggest that annotation accuracy alone does not fully characterize label quality. In weakly supervised settings, distributional properties and systematic biases also play critical roles. Although HVL does not explicitly correct label imbalance, its hybrid probability estimation method helps alleviate bias-related degradation, demonstrating strong adaptability.

## 3.4 FURTHER ANALYSES

**Training Cost Analysis.** To analyze the training cost of our method, we compare it with other weakly supervised approaches on the DTD and Caltech-101 datasets. As shown in Figure 2, HVL converges significantly faster, reaching high accuracy with lower computational cost. On DTD, HVL achieves approximately 67% accuracy much earlier than PLNL (around 65%) and SPMI (around 62%), which require more epochs and higher training costs. Meanwhile, PaPi and CPL show large fluctuations in accuracy and incur greater computational overhead.

Similar trends are observed on Caltech-101. HVL quickly reaches 94.77% accuracy with minimal variance and reduced training time, demonstrating strong efficiency and robustness. In contrast, DIRK converges more slowly to approximately 93%, consuming more training resources; PLNL and CPL stabilize around 94% and 93% accuracy respectively with higher costs, while SPMI and PaPi perform poorly in both accuracy and efficiency.

In summary, HVL exhibits superior training cost-efficiency, achieving high accuracy with less computation and more stable convergence compared to other methods.

**Influence of the Hyperparameter $\lambda$.** We investigate the sensitivity of our method to the hyperparameter $\lambda$ on EuroSAT and Caltech101. As shown in Figure 3, our method maintains consistently high accuracy across a wide range of $\lambda$ values. On EuroSAT, accuracy fluctuates slightly between

Table 3: Label accuracies by Qwen using three prompts and HVL classification performance on three datasets. Numbers in parentheses show improvements over Zero-shot LaFTer.

|  | DTD | EuroSAT | Caltech-101 |
|---|---|---|---|
| Prompt1 | 57.62 | 32.19 | 84.16 |
| HVL (prompt1) | 67.25 (+16.66) | 92.80 (+21.76) | 94.77 (+1.71) |
| Prompt2 | 59.79 | 46.56 | 74.39 |
| HVL (prompt2) | 67.49 (+16.90) | 90.21 (+19.17) | 94.89 (+1.83) |
| Prompt3 | 60.32 | 46.69 | 74.29 |
| HVL (prompt3) | 67.14 (+16.55) | 90.65 (+19.61) | 94.08 (+1.02) |

Table 4: Label accuracies from three VLMs (LLaVA, Qwen, CLIP) and HVL classification performance on three datasets. Numbers in parentheses show improvements over Zero-shot LaFTer.

|  | DTD | EuroSAT | Caltech-101 |
|---|---|---|---|
| Zero-shot LLaVA | 38.58 | 42.96 | 55.94 |
| HVL (LLaVA) | 60.99 (+10.40) | 80.81 (+9.77) | 94.35 (+1.29) |
| Zero-shot Qwen | 57.62 | 32.19 | 84.16 |
| HVL (Qwen) | 67.25 (+16.66) | 92.80 (+21.76) | 94.77 (+1.71) |
| Zero-shot CLIP | 43.48 | 39.02 | 92.27 |
| HVL (CLIP) | 63.06 (+12.47) | 91.42 (+20.38) | 96.39 (+3.33) |

91.46% and 92.80%, while on Caltech101 it stays above 94%, peaking at 94.77%. These results demonstrate that our method is robust to the choice of $\lambda$ and does not require extensive tuning to achieve strong performance.

**Influence of Prompt Variants.** To examine the impact of prompt design, we evaluate three variants under the Qwen model: a detailed Chinese prompt with step-by-step guidance, a simplified Chinese prompt, and an English translation of the original. As shown in Table 3, our method consistently outperforms the Zero-shot LaFTer baseline across all variants, demonstrating strong robustness to differences in prompt length, detail, and language. The full prompt texts are provided in the Appendix B.4.

**Influence of Different VLMs.** To assess the generalization ability of our method across different VLMs, we compare the label accuracy and HVL classification performance based on three VLMs: LLaVA (Liu et al., 2023), Qwen (Bai et al., 2025), and CLIP Radford et al. (2021). As shown in Table 4, regardless of the VLM used for initial label generation, human verification consistently yields substantial performance gains. These results highlight the robustness and adaptability of HVL to varying label qualities from diverse VLMs.

**Issue of Human Verification Rounds.** We examine the effect of different verification rounds (1, 2, and 3) on HVL performance. In each round, if a VLM-generated label is rejected by a human, a new label is generated until verification succeeds or the round limit is reached. As shown in Table 5, more rounds lead to higher label accuracy and improved classification performance, though the marginal gains diminish. On DTD, three rounds verification even outperforms FSFT.

Table 5: Classification accuracies on three datasets under different numbers of human verification rounds. Numbers in parentheses show improvements over Zero-shot LaFTer.

|  | DTD | EuroSAT | Stanford Cars |
|---|---|---|---|
| Verification ×1 | 57.62 | 32.19 | 44.30 |
| HVL (1 round) | 67.25 (+16.66) | 92.80 (+21.76) | 68.62 (+12.07) |
| Verification ×2 | 71.52 | 57.70 | 56.88 |
| HVL (2 rounds) | 71.57 (+20.98) | 94.95 (+23.91) | 73.26 (+16.71) |
| Verification ×3 | 78.62 | 63.32 | 65.68 |
| HVL (3 rounds) | 73.94 (+23.35) | 95.36 (+24.32) | 77.22 (+20.67) |

This trend occurs because early rounds help filter out obviously incorrect labels, while later rounds mainly verify semantically similar but suboptimal labels. Although the performance gains decrease, this multi-round verification remains crucial for ensuring label quality and model robustness.

**Influence of Erroneous Verification Rate.** To test the robustness of our method against erroneous verifications, we conducted experiments under varying rates of erroneous verifications. As shown in Figure 4, even with a significant increase in the rate of erroneous verifications, our method still maintains high performance. We attribute this robustness significantly to the role of the risk-consistent estimator. The incorporation of VLM knowledge effectively mitigates the noise introduced by erroneous verifications, enabling the model to maintain stable performance even under such conditions.

## 4 RELATED WORK

Weakly supervised learning (Zhou, 2018) greatly reduces annotation costs and has attracted growing interest. Common paradigms include semi-supervised learning (SSL) (Van Engelen & Hoos, 2020; Cao et al., 2021; Guo et al., 2022; Wei et al., 2024), partial-label learning (PLL) (Xia et al., 2023;

Liu et al., 2024; Wu et al., 2024), and complementary-label learning (CLL) (Wei et al., 2023; Ying et al., 2023; Li et al., 2025), which address different types of weak or incomplete supervision.

### 4.1 SEMI-SUPERVISED LEARNING (SSL).

SSL (Van Engelen & Hoos, 2020; Cao et al., 2021; Guo et al., 2022; Wei et al., 2024) assumes that only a limited portion of the training set is labeled, while the majority of samples in the training data remain unlabeled. It mainly includes entropy minimization methods, consistency regularization methods, and holistic methods. Entropy minimization aims to reduce the uncertainty of model predictions on unlabeled data, encouraging confident outputs (Grandvalet & Bengio, 2004; Lee, 2013). Consistency regularization promotes prediction stability under input perturbations, such as noise or transformations (Sajjadi et al., 2016; Tarvainen & Valpola, 2017; Miyato et al., 2019). Holistic methods, exemplified by MixMatch and FixMatch, integrate pseudo-labeling with data augmentation to fully exploit unlabeled data (Berthelot et al., 2019; Sohn et al., 2020). Despite reducing overall labeling costs, SSL still requires human annotators to select the correct label from a large candidate space, which can be non-trivial in practice.

### 4.2 PARTIAL-LABEL LEARNING (PLL).

In PLL, each training instance is annotated with a set of candidate labels, among which only one is correct but unspecified. Existing PLL methods are generally divided into two categories: identification-based approaches that infer the true label via iterative updates or discriminative models (Tang & Zhang, 2017; Tarvainen & Valpola, 2017; Xu et al., 2019) and average-based approaches that treat all candidate labels equally to handle label ambiguity (Cour et al., 2011; Hüllermeier & Beringer, 2006). Recent work has explored task-specific representation learning for better label disambiguation Xia et al. (2023), unified frameworks that leverage both partial-label and unlabeled data to ease supervision limitations (Liu et al., 2024), and knowledge distillation methods where a teacher model guides the student to better resolve label uncertainty (Wu et al., 2024). Nevertheless, noise within the candidate labels remains a challenge and must be addressed during training to prevent the model from learning incorrect patterns.

### 4.3 COMPLEMENTARY-LABEL LEARNING (CLL).

CLL (Wei et al., 2023; Ying et al., 2023; Li et al., 2025) assigns each sample a label indicating a class that it does not belong to, which reduces the difficulty of annotation while still enabling effective model training. Ishida et al. (2017) proposed an unbiased risk estimator based on complementary labels for multi-class classification. Subsequent works improved CLL by adding loss correction to reduce label noise (Yu et al., 2018), designing more robust loss functions (Gao & Zhang, 2021), and extending the method to more flexible scenarios (Ishida et al., 2019). However, the noisy and imperfect nature of complementary labels demands sophisticated methods, resulting in longer training times.

Despite significant progress in the aforementioned weak supervision approaches, they significantly neglect the interaction and collaboration between human annotators and VLMs, often resulting in low-quality supervision that lacks reliability.

## 5 CONCLUSION

In this paper, we propose a novel annotation setting called Human-Verified Label (HVL), which significantly improves label quality with limited human involvement. In this setting, humans are only required to verify labels generated by VLMs, greatly reducing the cost of human annotation. We theoretically derive a risk-consistent estimator to effectively extract supervisory signals from HVLs. Furthermore, we introduce a hybrid probability estimation method that integrates knowledge from both the VLM and the training model to more accurately estimate the conditional probability. Experimental results demonstrate the effectiveness and practicality of the proposed HVL setting across various datasets.

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

## A  The Use of Large Language Models (LLMs)

During the preparation of this manuscript, large language models (e.g., ChatGPT) were used solely for grammar checking, language polishing and enhancing readability. All initial drafts of the manuscript were written entirely by the authors. The authors carefully reviewed all AI-generated suggestions to ensure accuracy and academic rigor.

## B  Appendix

### B.1  Proof of Lemma 2

**Lemma 2.** Under the Condition 1, the conditional probabilities $p(y = i \mid x)$ can be expressed as a combination of the probability of human-verified true labels and the corrected probabilities of human-verified false labels as follows:

$$
p(y{=}i \mid x) = \underbrace{p(Y{=}[i,1] \mid x)}_{\text{Human-Verified True Labels}} + \underbrace{\sum_{\substack{j=1 \\ j \neq i}}^{K} p(y{=}i \mid Y{=}[j,0],\, x)\, p(Y{=}[j,0] \mid x)}_{\text{Human-Verified False Labels}} \tag{11}
$$

**Proof.** According to Condition 1, Bayes Rule and Total Probability Theorem,

$$p(y{=}i \mid x) = p(y{=}i, Y{=}[\hat{y}, 1] \mid x) + p(y{=}i, Y{=}[\hat{y}, 0] \mid x)$$

$$= \sum_{j=1}^{K} p(y{=}i, Y{=}[j, 1] \mid x) + \sum_{\substack{j=1 \\ j \neq i}}^{K} p(y{=}i, Y{=}[j, 0] \mid x)$$

$$= \sum_{j=1}^{K} p(y{=}i \mid Y{=}[j, 1], x) \, p(Y{=}[j, 1] \mid x) + \sum_{\substack{j=1 \\ j \neq i}}^{K} p(y{=}i \mid Y{=}[j, 0], x) \, p(Y{=}[j, 0] \mid x)$$

$$= p(y{=}i \mid Y{=}[i, 1], x) \, p(Y{=}[i, 1] \mid x) + \sum_{\substack{j=1 \\ j \neq i}}^{K} p(y{=}i \mid Y{=}[j, 0], x) \, p(Y{=}[j, 0] \mid x)$$

$$= p(Y{=}[i, 1] \mid x) + \sum_{\substack{j=1 \\ j \neq i}}^{K} p(y{=}i \mid Y{=}[j, 0], x) \, p(Y{=}[j, 0] \mid x) \tag{12}$$

## B.2 Proof of Theorem 3.

**Theorem 3.** To deal with the learning problem of HVLs, according to Condition 1 and Lemma 2, the classification risk $R_M(f)$ in Eq. (1) could be rewritten as:

$$R_{\mathrm{HVL}}(f) = \mathbb{E}_{p(x, Y=[\hat{y}, 0])} \, \bar{\mathcal{L}}(f(x), Y) + \mathbb{E}_{p(x, Y=[\hat{y}, 1])} \, \mathcal{L}(f(x), Y) \tag{13}$$

where $\bar{\mathcal{L}}(f(x), Y) = \sum_{\substack{i=1 \\ i \neq j}}^{K} p(y{=}i \mid Y{=}[j, 0], x) \, \mathcal{L}(f(x), Y)$

**Proof.** According to the Condition 1 and Lemma 2

$$R_{\mathrm{HVL}}(f) = \mathbb{E}_{x \sim p(x)} \sum_{i=1}^{K} p(y = i \mid x) \, \mathcal{L}(f(x), i)$$

$$= \mathbb{E}_{x \sim p(x)} \sum_{i=1}^{K} \sum_{\substack{j=1 \\ j \neq i}}^{K} p(y{=}i \mid Y{=}[j, 0], x) \, p(Y{=}[j, 0] \mid x) \mathcal{L}(f(x), Y)$$

$$+ \mathbb{E}_{x \sim p(x)} \sum_{i=1}^{K} p(Y{=}[i, 1] \mid x) \mathcal{L}(f(x), Y)$$

$$= \mathbb{E}_{x \sim p(x)} \sum_{j=1}^{K} p(Y{=}[j, 0] \mid x) \sum_{\substack{i=1 \\ i \neq j}}^{K} p(y{=}i \mid Y{=}[j, 0], x) \mathcal{L}(f(x), Y) \tag{14}$$

$$+ \mathbb{E}_{x \sim p(x)} \sum_{i=1}^{K} p(Y{=}[i, 1] \mid x) \mathcal{L}(f(x), Y)$$

$$= \mathbb{E}_{p(x, Y=[\hat{y}, 0])} \sum_{\substack{i=1 \\ i \neq j}}^{K} p(y{=}i \mid Y{=}[j, 0], x) \mathcal{L}(f(x), Y) + \mathbb{E}_{p(x, Y=[\hat{y}, 1])} \mathcal{L}(f(x), Y)$$

$$= \mathbb{E}_{p(x, Y=[\hat{y}, 0])} \, \bar{\mathcal{L}}(f(x), Y) + \mathbb{E}_{p(x, Y=[\hat{y}, 1])} \, \mathcal{L}(f(x), Y)$$

## B.3 The Details of Datasets

In this section, we provide a detailed description of datasets used in our experiments.

- CIFAR-100: A coarse-grained dataset comprising 60,000 color images divided into 100 classes. Each image is given in a $32 \times 32 \times 3$ format, and each class contains 500 training images and 100 test images.

- Caltech-101: A coarse-grained dataset comprises images from 101 object categories and a background category that contains the images not from the 101 object categories. Each object category contains approximately 40 to 800 images, with most classes having about 50 images. The image resolution is approximately $300 \times 200$ pixels.

- Food-101: A fine-grained dataset in the food domain, comprising 101,000 images divided into 101 food categories. Each class contains 500 training images and 300 test images. The labels for the test images have been manually cleaned, while the training set contains some noise.

- Stanford Cars: A fine-grained dataset in the car domain, comprising 16,185 images categorized into 196 car classes. The training set contains 6,509 images, and the test set contains 8,041 images. Categories are typically at the level of Make, Model, Year. The images are $360 \times 240$ pixels.

- DTD : A coarse-grained dataset focused on describable textures, comprising 5,640 images categorized into 47 texture classes such as "striped", "dotted", and "zigzagged". Each class contains 120 images with significant intra-class variability. The images are collected in the wild and vary in resolution and aspect ratio.

- EuroSAT: A coarse-grained dataset for land use and land cover classification, consisting of 27,000 Sentinel-2 satellite images covering 10 classes such as "Forest", "River" and "Residential". Each class contains 2,000–3,000 RGB images with a resolution of $64 \times 64$ pixels. The dataset exhibits high inter-class visual variability due to differing geographical patterns.

### B.4 THE FULL PROMPT TEXTS

In this section, we provide the full prompt texts used in our experiments.

Prompt 1 and Prompt 3 share the same semantic structure but are written in Chinese and English respectively. The prompt consists of the following instructions:

- Step 1: Carefully analyze the main object and details in the image.
- Step 2: Compare the image with the following candidate labels and select the best match.
- Step 3: Output only the numeric index of the most suitable label, without any additional text, punctuation, or explanation.

Prompt 2 is a simplified version in Chinese:

Analyze the image and select the most relevant option. Only output the index. Do not explain or include the category name.

