# OpenReview forum: "Leveraging Human Collaboration: Learning from Human-Verified Labels"
_ICLR.cc/2026/Conference — ICLR 2026 Conference Withdrawn Submission_

### Official Review · Reviewer_8voA · 2025-10-26

**Soundness:** 2
**Presentation:** 2
**Contribution:** 2
**Rating:** 2
**Confidence:** 5

**Summary:**

The paper identifies that VLM-generated labels are noisy and hurt classifier performance. They propose Human-Verified Labels (HVLs), a new annotation method where the Vision-Language Models (VLMs) like CLIP generates an initial label for each image. A human annotator verifies whether that label is correct (True/False). A classifier learns from these binary verification signals. Adding human verification signals improves learning from noisy VLM labels.

**Strengths:**

1. The paper identifies a problem in VLMs and MLLMs such as CLIP/Qwen, where they make incorrect predictions and show that with Human corrections they can make the prediction robust.

2. The experiments are extensively evaluated on 6 diverse datasets and multiple VLMs (CLIP, Qwen, LLaVA).

3. The probability estimation method of combining VLM and model predictions is reasonable.

**Weaknesses:**

1. Given that the study is conducted on single-label settings (where there is only one object in the image), it would be more efficient to directly ask humans to label the object, instead of running a VLM and then having humans check only the one label that the VLM generates.

2. When the output of the VLM is wrong, the human would only indicate that it is incorrect, we would not obtain the correct label of the object. The cost of labeling it as incorrect would be the same as directly labeling the correct object by looking at the image (because it is a single-label setting).

3. A minor weakness is that the assumption that human predictions are always correct needs to be discussed further. There are cases of mislabeled or missing labels in popular datasets; sometimes this happens due to difficulty in recognizing the images (e.g., the CIFAR-10 dataset). A discussion on this would help strengthen the paper.

4. I believe that using HVLs would provide more meaningful contribution in Multi-label settings (image contains multiple objects, we need to identify all of them) or specialized domains like medical imaging

**Questions:**

Please see the weaknesses section.

---

### Official Review · Reviewer_FADN · 2025-10-28

**Soundness:** 3
**Presentation:** 3
**Contribution:** 1
**Rating:** 4
**Confidence:** 5

**Summary:**

This work tackles a key limitation of pre-trained Vision-Language Models (VLMs): while VLMs lower annotation costs via automated label generation, their noisy labels (without human input) degrade classifier performance. To address this, the authors propose Human-Verified Labels (HVLs)—a novel annotation setting integrating limited human collaboration. Each HVL requires only simple correctness judgments from human. Methodologically, the authors derive a risk-consistent estimator to leverage correlations between VLM-label distributions and human verification, plus a hybrid probability method fusing VLM and model knowledge to improve conditional probability accuracy.

**Strengths:**

The authors propose a novel framework—Human-Verified Labels (HVL)—as well as a theoretically grounded method, and carry out comprehensive experiments.

**Weaknesses:**

1. From a personal perspective, the HVL setting seems somewhat counterintuitive. If human verification is to be utilized, it is unclear why annotators are not directly asked to label samples with ground-truth labels. For instance, the Food-101 dataset contains over 100k samples; in this case, the efficiency of manually verifying VLM-generated labels is clearly inferior to that of manually labeling 4–8 samples per class.

2. In terms of performance, the experimental results presented in this paper do not show obvious advantages over few-shot fine-tuning methods on some datasets.

**Questions:**

Please respond to the weaknesses mentioned above.

---

### Official Review · Reviewer_rsB7 · 2025-10-30

**Soundness:** 3
**Presentation:** 3
**Contribution:** 3
**Rating:** 4
**Confidence:** 4

**Summary:**

The paper introduces Human-Verified Labels (HVLs), a hybrid annotation scheme where labels produced by vision–language models (VLMs) (e.g., CLIP, Qwen) are verified by humans with a binary signal (h ∈ {0,1}) indicating whether the proposed label is correct.
The authors derive a risk-consistent estimator that rewrites the classification risk in terms of observed HVLs and unknown conditional terms p(y=i | Y=[j,0],x). They propose a hybrid conditional probability estimator that fuses VLM softmax scores and the model’s own softmax via a hyperparameter λ.

**Strengths:**

1. Verifying a single candidate label is cheaper than fully labeling; the HVL concept captures a pragmatic human–AI collaboration regime that practitioners can adopt.

2. The risk re-writing (Lemma 2, Theorem 3) connects observable HVL quantities to the true risk and motivates learning from both human-verified true and false labels. The derivations are straightforward and useful for designing loss terms.

3. The hybrid estimator (P = λ P̂_VLM + (1−λ) P̄_model) is easy to implement, and experiments indicate robustness to λ.

**Weaknesses:**

1. The HVL Condition (Def. 1) assumes that when humans verify a VLM label as true, it is *perfectly* the ground truth: p(y=i | Y=[i,1],x)=1. In practice, human verification can be mistaken (ambiguity, low annotator expertise). While the paper experiments with an “erroneous verification rate” (Figure 4), the theoretical results rely on the ideal condition—there is limited theoretical or empirical treatment of systematic misverification (e.g., consistent false positives). This weakens the theoretical guarantees.

2. The central claim is “limited human effort,” but the manuscript lacks concrete numbers for annotation time per verification, inter-annotator agreement, cost per instance, and realistic crowdsourcing experiments. The only related analysis is verification rounds (Table 5) and a synthetic erroneous verification rate plot (Figure 4), which do not substitute for live human studies.

3. The conditional probability p(y=i | Y=[j,0],x) is estimated heuristically. The hybrid fusion is practical, but the paper lacks statistical analysis of estimator bias/variance or a principled calibration strategy. It is unclear how well the fused distribution approximates the true conditional used in the risk re-writing, especially early in training when model predictions are poor.

4. The paper compares to weakly supervised and automatic labeling methods, but omits direct comparisons to: (a) Active learning strategies that query human labels selectively, or (b) Simple verification baselines (e.g., accept VLM label if confidence > threshold; otherwise ask human). These are natural, low-cost baselines for a human-verified setup.

**Questions:**

1. In practice, human verification is noisy. Have you measured real human verification error rates (false accept / false reject)? If so, please report them and indicate how they align with the synthetic erroneous verification experiments in Figure 4.
2. For the multi-round verification (Table 5): how are new candidate labels generated after a rejection? Is it the next top-1 from the same VLM, re-sampled with prompts, or from a different VLM? Please state the exact algorithm and the distribution of rounds used.
3. How sensitive are results to the choice of the base model (LaFTer) and backbone (CLIP-ViT-B/32)? Would HVL gains persist with stronger backbones or different fine-tuning recipes?

---

### Official Review · Reviewer_b1CD · 2025-11-01

**Soundness:** 2
**Presentation:** 2
**Contribution:** 2
**Rating:** 2
**Confidence:** 3

**Summary:**

This paper introduces Human-Verified Labels (HVLs) — a novel annotation paradigm that integrates limited human collaboration with Vision-Language Model (VLM)-generated labels. Additionally, a risk estimator is proposed to explpre and leverage human verification information. Experiments demonstrate its effectiveness.

**Strengths:**

1. The authors propose to model learning with human collaboration, which might provides more effective information while requiring less cost.
2. The authors provide some theoretical analysis for their risk estimator when introducing the extra human information.

**Weaknesses:**

1. The process of introducing human-verified labels in practical experiments is not clearly described. It remains ambiguous whether the verification was conducted by actual annotators or simulated using ground-truth labels. If the latter, the method effectively uses oracle-level information that is unavailable to the baseline methods, which raises fairness concerns in comparison.

2. More baselines. Recent works in pseudo-label denoising and self-training (e.g., Debiased Learning from Naturally Imbalanced Pseudo-labels, Probabilistic Label Correction) could serve as strong baselines. Their absence makes it difficult to assess whether human verification yields qualitatively different benefits compared to automated noise filtering.

3. The theoretical analysis, while correct and complete, does not introduce substantially new ideas beyond the existing frameworks in partial-label learning or complementary-label learning. The proposed “risk-consistent estimator” closely parallels prior formulations such as Provably Consistent Partial-Label Learning and Single-Positive Multi-Label Learning with Label Enhancement. As a result, the originality of the theoretical contribution is limited, and the main innovation of the paper lies more in the problem formulation and empirical validation rather than the theory itself.

4. Writing typo. The format of subsection is inconsistent. (some subsection contains the comma. see Sec 4.1).

**Questions:**

See Weaknesses.

---

### Note · Authors · 2025-11-30

**Comment:**

After discussion among the authors, it was decided to withdraw the manuscript.

**Withdrawal Confirmation:**

I have read and agree with the venue's withdrawal policy on behalf of myself and my co-authors.